# Global HIV prevention, treatment, and care interventions and strategies for key populations: Protocol for a scoping review

Emmanuel Kumah[1]*, Dorothy Serwaa Boakye[1], Eunice Agyei[2], Richard Boateng[1], Veronica Penaman Asamoah[1], Emmanuel Osei Tutu[1]

1 Department of Health Administration and Education, Faculty of Health, Allied Sciences and Home Economics Education, University of Education, Winneba, Ghana, 2 General Services Directorate, Komfo Anokye Teaching Hospital, Kumasi, Ghana

☯ These authors contributed equally to this work.
* ababiohemmanuel@gmail.com

## Abstract

### Introduction

Since its emergence, HIV/AIDS has remained one of the most significant global health challenges, with key populations—such as sex workers, men who have sex with men (MSM), transgender people, people who inject drugs (PWID), and individuals in prisons or other closed settings—disproportionately bearing the burden of the epidemic. These groups, often at heightened risk due to social, legal, and structural vulnerabilities, face persistent barriers to accessing prevention, treatment, and care services. Despite progress in reducing new infections and improving treatment outcomes, these disparities, exacerbated by stigma, structural inequalities, and insufficient political commitment, continue to limit the effectiveness of global HIV responses.

### Aim

This scoping review protocol aims to systematically map the range of HIV prevention, treatment, and care interventions and strategies targeting key populations worldwide. Rather than formally evaluating effectiveness, the review will describe the nature, extent, and types of interventions implemented, identify barriers to implementation, and highlight gaps in research and practice.

### Methods

Following the Joanna Briggs Institute (JBI) guidelines for scoping reviews, the study will systematically identify and analyze evidence from multiple databases, including PubMed, Embase, CINAHL, Scopus, and PsycINFO, alongside regional and grey literature sources. The review will include studies focusing on key populations and

**Data availability statement:** No datasets were generated or analysed during the current study. All relevant data from this study will be made available upon study completion.

**Funding:** The author(s) received no specific funding for this work.

**Competing interests:** The authors have declared that no competing interests exist.

evidence-based interventions, such as prevention tools, treatment strategies, and policy or structural interventions. Data will be extracted and synthesized using quantitative and qualitative approaches, with results presented through descriptive statistics and thematic analysis. Findings will inform the development of a comprehensive, evidence-based framework tailored to the unique needs of key populations.

## Conclusion

By mapping available interventions and strategies for HIV prevention, treatment, and care among key populations, this review will provide a comprehensive overview of existing approaches, barriers, and gaps. The findings will inform future research, policy, and practice, supporting more targeted, inclusive, and sustainable HIV responses that contribute to global efforts to end AIDS as a public health threat by 2030.

## Introduction

Since its emergence in 1981, HIV, the virus that causes AIDS, has become one of the most significant global health and development challenges. To date, approximately 88 million people have contracted HIV, and tens of millions have died from AIDS-related causes [1]. Currently, around 40 million people live with HIV globally, with sub-Saharan Africa accounting for two-thirds of this population, making it the most severely affected region. Other regions, including Asia, the Pacific, Latin America, and Eastern Europe, also face substantial burdens. These regions often grapple with additional challenges such as infectious diseases, food insecurity, and other pressing health and development issues [1].

Over the decades, global efforts have sought to address the epidemic. Initial international responses included the creation of the World Health Organization's (WHO) Global Program on AIDS in 1987, followed by the establishment of the Joint United Nations Program on HIV/AIDS (UNAIDS) in 1996 and the Global Fund to Fight AIDS, Tuberculosis, and Malaria in 2001. These initiatives, combined with the contributions of affected countries and civil society, have played a pivotal role in advancing the global response [2,3]. Central to these efforts are the Sustainable Development Goals (SDGs), adopted in 2015, which aim to end AIDS as a public health threat by 2030. Progress toward these goals has been significant, with annual new infections dropping by 60% since 1995 and AIDS-related deaths declining by 69% since 2004. Additionally, 86% of people living with HIV were aware of their status in 2023, with 77% accessing treatment and 72% achieving viral suppression [1].

Despite this progress, substantial gaps remain in achieving global HIV prevention and treatment targets. Disparities in access to services persist across regions and population groups, with key populations facing the greatest challenges [4–6]. The WHO defines key populations as those at heightened risk for HIV due to social, legal, and structural vulnerabilities. These groups include sex workers, men who have sex with men (MSM), transgender people, people who inject drugs (PWID), and individuals in prisons or other closed settings [7]. They play a critical role in HIV transmission

dynamics and are essential partners in effective responses. However, in many countries, prevention and treatment services for key populations remain inadequate, leaving these groups disproportionately affected [8].

Key populations bear a significantly higher HIV burden than the general population. In 2023, while global HIV prevalence among adults aged 15–49 was 0.8%, prevalence among MSM, transgender people, PWID, and sex workers ranged from 3% to 9.2% [1]. Alarmingly, 55% of all new infections in 2022 were among key populations and their partners, up from 44% in 2010 [9]. Despite the availability of prevention tools such as condoms and pre-exposure prophylaxis (PrEP), access remains limited for these vulnerable groups, particularly in low- and middle-income countries. Similarly, harm-reduction services for PWID are often absent or insufficient [10]. Expanding access to these prevention tools in high-incidence regions could dramatically reduce new infections.

Treatment coverage for key populations remains significantly below global averages, highlighting persistent disparities in healthcare access and outcomes [11]. While global treatment efforts have markedly improved overall coverage—30.7 million of the estimated 39.9 million people living with HIV were receiving antiretroviral therapy (ART) in 2023 [1]—key populations continue to face substantial barriers that limit their access to and benefits from these services. For example, recent data from sub-Saharan Africa reveal that in countries with 80% ART coverage in the general population, coverage among female sex workers and gay men or other men who have sex with men was 11–13% lower, and 30% lower among transgender women [12]. These disparities are driven by a combination of structural inequalities, pervasive stigma, and discrimination, as well as entrenched social and legal challenges that disproportionately impact marginalized groups [13–15]. A 2023 survey across 42 countries found that nearly half of respondents expressed discriminatory attitudes toward people living with HIV [16]. Such stigma frequently permeates healthcare settings, exacerbating inequities in access to care and leading to poorer health outcomes for these vulnerable populations [13,15].

Criminalization and punitive laws further compound these challenges. In many countries, same-sex relations, sex work, and drug use remain criminalized, exposing key populations to harassment, arrest, and violence [16]. Such legal barriers discourage individuals from seeking HIV prevention and treatment services, disrupt community-led outreach, and reinforce stigma and discrimination [17,18]. For instance, available evidence indicates that in Tanzania, only 49% of estimated key populations access health services, with laws criminalizing drug use, same-sex relations, and sex work leading to harassment and arrests that further marginalize these groups [19]. Laws that criminalize HIV transmission, non-disclosure, or exposure also create fear and mistrust, undermining effective public health responses. These legal and policy environments not only restrict access to essential services but also perpetuate structural inequalities that intensify the disproportionate burden of HIV among key populations [13–15,17,18].

Social inequalities, gender-based violence, and insufficient political commitment exacerbate these challenges [17,18]. Many governments lack the political will to fund comprehensive prevention programs for key populations, relying instead on underfunded community-led organizations to fill critical gaps. As a result, at least half of all key populations are not reached by prevention services [20].

The projected trajectory of the HIV epidemic underscores the urgency of scaling up effective interventions. Without significant changes, UNAIDS estimates that 46 million people could be living with HIV by 2050. Even if ambitious targets are met, nearly 30 million people will require lifelong care [20]. With no universally accessible vaccine or cure, the global community faces an ongoing public health challenge.

Given the disproportionate burden of HIV among key populations, this scoping review aims to synthesize evidence on effective strategies and interventions to address their unique needs. A scoping review is particularly appropriate because it allows for the systematic mapping of a broad and heterogeneous body of evidence, encompassing diverse interventions, study designs, and contexts [21]. This approach is more suitable than a traditional systematic review, which typically focuses on narrowly defined questions and formal effectiveness evaluations [21]. By identifying successful approaches and examining barriers to service access, the study seeks to inform policies and programs that enhance HIV outcomes and promote equity. The culmination of this research will be the development of an integrated framework for HIV

prevention, treatment, and care tailored specifically to key populations. This framework will bring together comprehensive strategies for prevention, treatment, and ongoing care, emphasizing a holistic and equitable approach. By integrating evidence-based practices and addressing systemic barriers, it will provide a blueprint for creating inclusive and impactful HIV programs. Ultimately, this integrated approach aims to improve health outcomes, reduce disparities, and contribute to a more effective and just global HIV response, ensuring no one is left behind in the fight against HIV/AIDS.

### Review objectives

The main objectives of this systematic scoping review are to:

1. Identify and analyze evidence-based interventions and strategies for HIV prevention and treatment that have demonstrated success in key populations;

2. Explore the structural, social, and individual barriers preventing successful implementation of these interventions and strategies;

3. Examine gaps in policy, research, and practice, highlighting areas that require further attention to optimize HIV response strategies; and

4. Provide actionable recommendations for policymakers, healthcare providers, and stakeholders, and, based on the findings, develop an integrated framework for HIV prevention, treatment, and care tailored to key populations to ensure a comprehensive and equitable response.

## Methods

### Design

This scoping review will follow the Joanna Briggs Institute (JBI) guidelines for scoping reviews, which outline seven systematic stages: (1) defining the research question, (2) developing the protocol, (3) applying the PCC (Population/Concept/Context) framework, (4) conducting systematic searches, (5) screening studies, (6) extracting and charting relevant data, and (7) synthesizing and reporting the evidence [22]. The protocol will be reported in alignment with the Preferred Reporting Items for Systematic Reviews and Meta-Analysis Protocols (PRISMA-P) (S1 Appendix) [23] and the PRISMA Extension for Scoping Reviews (ScR) guidelines (S2 Appendix) [24]. The protocol has been registered with the Open Science Framework (https://doi.org/10.17605/OSF.IO/5S92C).

### Research question

The review will be guided by the following research question: What are the effective interventions and strategies for HIV prevention and treatment among key populations, and what barriers hinder their successful implementation?

### Search strategy

We will employ a three-step search strategy to ensure a comprehensive and systematic review. In the first phase, we will search PubMed to identify and expand keywords related to HIV prevention, HIV treatment, and key populations (Table 1) by reviewing titles and abstracts. Based on this initial search, the strategy will be refined to include all relevant index terms and keywords identified from the titles and abstracts of retrieved articles. In the second phase, the search strategy will be tailored and applied to multiple information sources, including PubMed, Embase, CINAHL, Scopus, PsycINFO, and the Cochrane Library. Additional regional and context-specific databases, such as African Journals Online (AJOL), Latin American and Caribbean Health Sciences Literature (LILACS), and the WHO Global Index Medicus, will also be searched for relevant publications. The third phase will involve a manual search to identify any additional relevant literature. This will

**Table 1. Search strategy for initial search on PubMed.**

| No. | Keywords | Search Terms |
| --- | --- | --- |
| 1 | HIV prevention | HIV prevention OR protection OR awareness program OR transmission control OR "public health campaign" |
| 2 | HIV treatment | HIV treatment OR therapy OR medication OR clinical care OR healthcare access OR management |
| 3 | Interventions/strategies | Intervention OR strategy |
| 4 | Key populations | Key population OR "men who have sex with men" OR transgender people OR "people who inject drugs" OR "individuals in prisons" OR sex workers |
| 5 | | 1-4 combined with "AND" |

include reviewing the reference lists of all studies included in the review, using Google Scholar to locate grey literature, and consulting the WHO Global Health Library and AIDSinfo/UNAIDS for HIV/AIDS-related literature, policies, guidelines, and reports on key populations.

## Definition of the inclusion and exclusion criteria

The inclusion and exclusion criteria for this review have been defined following JBI guidelines, using the PCC approach, by considering population/participants, concept, context, and evidence types.

## Population

The population of interest includes key populations as defined by the World Health Organization (WHO). These groups encompass sex workers, regardless of gender; men who have sex with men (MSM); transgender people; people who inject drugs (PWID); and individuals in prisons or other closed settings. Additionally, studies involving partners of key populations or healthcare providers working with these groups will be included if they directly address HIV interventions for key populations. Studies focusing solely on the general population without specific reference to key populations or addressing other vulnerable groups not categorized as key populations by the WHO will be excluded.

## Concept

The focus of the review is on evidence-based interventions and strategies for HIV prevention, treatment, and care specifically targeted at key populations. These may include prevention measures such as condom promotion, pre-exposure prophylaxis (PrEP), needle exchange programs, harm-reduction initiatives, HIV education, and community-based outreach programs. Treatment and care interventions, including antiretroviral therapy (ART), linkage to care, retention strategies, mental health services, and peer support programs, will also be considered. Policy and structural interventions, such as decriminalization of key population activities, stigma reduction, healthcare provider training, and efforts to ensure equitable healthcare access, are also within the scope of this review. Interventions unrelated to HIV prevention, treatment, or care, as well as strategies targeting other diseases or conditions, even if studied in key populations, will be excluded.

## Context

The review will include studies conducted globally, with particular attention to low- and middle-income countries, such as those in sub-Saharan Africa, Asia, the Pacific, Latin America, and Eastern Europe, where the burden of HIV is highest and structural barriers are significant. High-income countries where key populations face disparities in access to services are also included. Studies conducted in diverse settings, such as healthcare facilities, community-based environments, prisons, or other closed settings, will be considered. However, studies focused on general population settings without disaggregated data for key populations, as well as research not addressing HIV prevention, treatment, or care services, will be excluded.

## Evidence types

The review will include primary research studies (qualitative, quantitative, and mixed-method studies), systematic reviews, scoping reviews, and meta-analyses that synthesize evidence on HIV interventions for key populations. Policy and program evaluations related to HIV prevention and treatment for key populations will also be included. Reports, policy briefs, guidelines, and conference proceedings containing unpublished relevant data will be considered. Given the variable quality of program reports and grey literature, such documents will not be excluded a priori but will be carefully appraised and contextualized during analysis. Their contribution will be considered primarily in terms of providing practical insights, complementing findings from peer-reviewed studies, and highlighting implementation experiences. Any limitations in methodological rigor or reporting quality will be acknowledged explicitly to ensure transparency in interpreting their findings.

Exclusions will include opinion pieces, editorials, and commentaries lacking relevant data. Studies published in languages other than English will also be excluded. This decision is based on practical considerations, including limitations in translation resources and the need to ensure consistency and accuracy in data extraction and synthesis. However, the exclusion of non-English studies may introduce a potential language bias, as relevant findings published in other languages might not be captured. This limitation will be acknowledged in the review, and its possible influence on the comprehensiveness of the evidence base will be discussed in the interpretation of results.

## Study selection

All studies identified through the electronic database search will be exported to EndNote for duplicate removal. The screening process will then proceed in two stages. In the first stage, titles and abstracts will be independently assessed by two researchers to exclude studies that do not meet the inclusion criteria. The second stage will involve a detailed review of the full texts of the remaining articles to further refine the selection. Each stage of the study selection process will be conducted independently by the two researchers and cross-checked to ensure accuracy and consistency. Any discrepancies will be resolved through discussion, and if consensus cannot be reached, a third researcher will be consulted for final adjudication. The entire study selection process will be systematically documented using the PRISMA flow diagram (Fig 1) to ensure transparency and reproducibility.

## Data extraction

Data extraction for this scoping review will be systematically conducted to ensure the thorough and accurate capture of relevant information. Two independent reviewers will extract data using a standardized Microsoft Excel spreadsheet (S3 Appendix), predesigned to capture key items such as article information (author(s), publication year, and journal), study characteristics (objectives, geographical and contextual setting, study design, and sample characteristics), intervention details (type of intervention or strategy and targeted key population(s)), and outcomes and impact (reported outcomes, factors influencing effectiveness, and barriers to implementation). The extraction will also consider contextual factors, including social, structural, and individual barriers, as well as enabling factors and facilitators, along with policy and structural aspects, such as policy interventions (e.g., stigma reduction, decriminalization, or healthcare reforms) and structural interventions (e.g., harm reduction or provider training). The extraction tool will be iteratively refined throughout the data collection process to ensure the inclusion of all critical information, with any modifications documented in the final scoping review.

## Data analysis and synthesis

Quantitative data will be summarized using descriptive statistics to capture trends and patterns across the studies, including prevalence rates, intervention coverage, and effectiveness measures presented using metrics such as means, medians, standard deviations, effect sizes, and ranges. Tables will be used to organize and present data on intervention types, target populations, geographic distribution, and reported outcomes, providing a broad understanding of intervention reach and outcomes across different settings and populations.

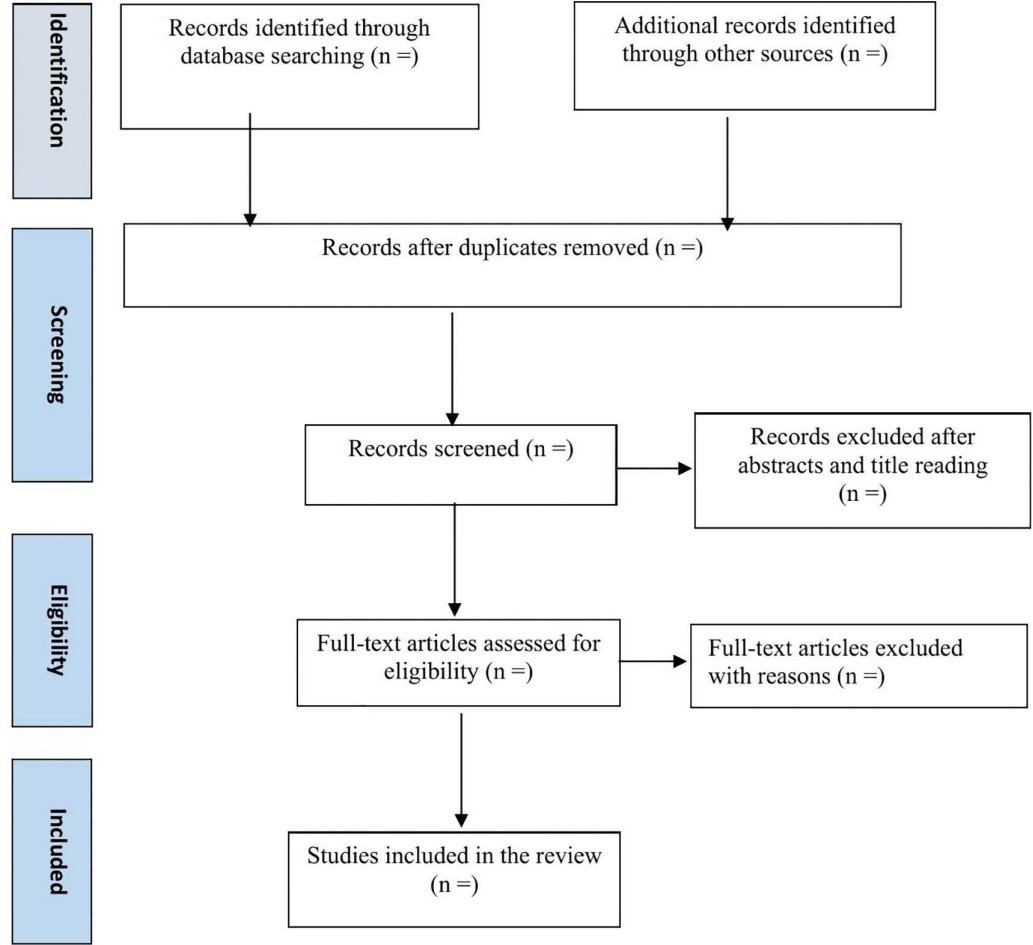

**Fig 1. Flow chart of study selection.**

Qualitative data will be analyzed through systematic content analysis to explore barriers preventing the successful implementation of HIV prevention and treatment strategies for key populations. This process will involve coding to identify recurring themes related to structural, social, and individual barriers, followed by thematic analysis to categorize these barriers into actionable insights.

Integration of findings will occur through a convergent synthesis process in which quantitative descriptive results (e.g., coverage levels, outcome trends) will be compared and mapped alongside qualitative themes (e.g., barriers, facilitators, contextual factors). This will allow the review to highlight not only the extent and distribution of interventions but also the contextual factors shaping their effectiveness. For example, descriptive data on low intervention coverage in certain regions will be interpreted together with qualitative evidence on stigma, criminalization, or healthcare system barriers.

The final framework will synthesize these complementary strands of evidence, linking quantitative patterns with qualitative explanations to demonstrate their combined impact on intervention success. Based on this integrated analysis, actionable strategies such as targeted policies, improved resource allocation, stigma reduction, and community-driven approaches will be proposed. The synthesis process will be iterative and transparent, ensuring alignment with the review objectives and accurately reflecting the current evidence base.

### Ethical consideration

This study does not require formal ethical approval, as it will rely solely on published data.

### Timeline

The study is scheduled to commence in June 2025 with the literature search. A first draft is expected by April 2026, and the final draft is targeted for submission to a journal by June 2026.

### Limitations

This review has several limitations. First, it will not assess the methodological rigor of the included studies, as the purpose of a scoping review is to map the evidence base rather than evaluate study quality. Consequently, this may limit the ability to make definitive practice recommendations. Second, while the inclusion of grey literature broadens the scope to capture relevant but unpublished or non–peer-reviewed evidence, it may also introduce variability in the quality and reliability of findings. Third, the review will only include studies published in English, which may exclude relevant evidence in other languages and lead to the omission of region-specific or culturally nuanced interventions and strategies. Fourth, structural stigma and criminalization faced by key populations may constrain what is studied, published, or disseminated, particularly in regions where such groups are marginalized or criminalized. This could result in gaps or biases in the available evidence base. To address this, the synthesis will interpret findings with sensitivity to the ways stigma and structural barriers shape both the availability and the framing of evidence. Finally, despite the comprehensive search strategy, some relevant literature may still be missed due to variations in indexing practices across databases or limitations in search terms. Collectively, these factors may influence the breadth, depth, and generalizability of the review findings.

## Discussion and conclusion

Despite significant progress in global HIV response efforts, key populations remain underserved, with limited access to evidence-based interventions and pervasive challenges such as stigma, discrimination, and policy-related barriers [4–6]. This review aims to address critical gaps in the literature by synthesizing evidence on effective interventions and identifying barriers to their implementation. By adopting the Joanna Briggs Institute methodology and leveraging a robust, multi-step search strategy, the review will ensure comprehensive coverage of the evidence base. The inclusion of both quantitative and qualitative studies, as well as policy and program evaluations, will allow for a holistic analysis of interventions and their contextual influences. Furthermore, the integration of findings through a mixed-methods synthesis will provide a more refined understanding of the interplay between intervention efficacy and structural barriers, informing the development of an integrated framework for HIV prevention, treatment, and care.

The findings of this study will contribute to a more inclusive and effective global HIV response by addressing the unique needs of key populations. By synthesizing evidence and providing actionable insights, this review will empower stakeholders to design and implement equitable HIV prevention, treatment, and care strategies. Ultimately, this work aligns with global efforts to achieve the Sustainable Development Goal of ending AIDS as a public health threat by 2030, ensuring no one is left behind.

To maximize its impact, the findings of this review will be disseminated through multiple channels targeting diverse audiences. Academic publication in an open access peer-reviewed journal will ensure that researchers, clinicians, and policymakers can access the findings. Results will also be presented at international and regional conferences on HIV, public health, and policy. Summarized findings will be shared through policy briefs and reports with government agencies, non-governmental organizations, and community-based organizations working with key populations. Tailored communication materials will be developed to share findings with affected communities, fostering awareness and advocacy. Dissemination will also include active engagement with key population networks to validate findings and promote uptake

at the community level, ensuring that the voices of those most affected are central to interpreting and applying the results. Additionally, online platforms, including social media, webinars, and blogs, will be utilized to engage a broader audience and stimulate dialogue on the outcomes of the review.

## Supporting information

**S1 Appendix. PRISMA checklist.**
(DOCX)

**S2 Appendix. The PRISMA Extension for Scoping Reviews (ScR) guidelines.**
(DOCX)

**S3 Appendix. Data extraction tool for the scoping review.**
(DOCX)

## Author contributions

**Conceptualization:** Emmanuel Kumah.

**Data curation:** Dorothy Serwaa Boakye.

**Formal analysis:** Emmanuel Kumah, Dorothy Serwaa Boakye.

**Investigation:** Emmanuel Kumah, Dorothy Serwaa Boakye, Richard Boateng.

**Methodology:** Emmanuel Kumah, Dorothy Serwaa Boakye, Eunice Agyei, Richard Boateng.

**Project administration:** Emmanuel Kumah.

**Resources:** Emmanuel Kumah, Dorothy Serwaa Boakye, Eunice Agyei, Richard Boateng, Veronica Penaman Asamoah.

**Software:** Emmanuel Osei Tutu.

**Supervision:** Emmanuel Kumah.

**Validation:** Emmanuel Kumah.

**Visualization:** Emmanuel Kumah.

**Writing – original draft:** Emmanuel Kumah, Dorothy Serwaa Boakye, Eunice Agyei, Richard Boateng, Veronica Penaman Asamoah, Emmanuel Osei Tutu.

**Writing – review & editing:** Emmanuel Kumah, Dorothy Serwaa Boakye, Eunice Agyei, Richard Boateng, Veronica Penaman Asamoah, Emmanuel Osei Tutu.

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
