## [Decision Letter · Decision Letter 0]

23 Sep 2025

Dear Dr. Kumah,

Thank you for submitting your manuscript to PLOS ONE. After careful consideration, we feel that it has merit but does not fully meet PLOS ONE’s publication criteria as it currently stands. Therefore, we invite you to submit a revised version of the manuscript that addresses the points raised during the review process.

We look forward to receiving your revised manuscript.

Kind regards,

Udoka Okpalauwaekwe, MD, MPH, PhD

Academic Editor

PLOS ONE

Journal Requirements:

3. Please upload a new copy of Figure 1 as the detail is not clear. Please follow the link for more information: https://blogs.plos.org/plos/2019/06/looking-good-tips-for-creating-your-plos-figures-graphics/

Additional Editor Comments:

This protocol addresses a highly relevant global health issue by proposing a systematic scoping review of HIV prevention, treatment, and care interventions for key populations. The rationale is strong, the methodology is anchored in JBI and PRISMA-ScR guidelines, and the inclusion of both biomedical and regional databases enhances comprehensiveness. The strengths lie in its holistic framing and attention to structural barriers. However, there are notable limitations: the scope risks overreaching by conflating mapping with effectiveness assessment; restricting to English-only studies may introduce language bias also; the integration strategy for quantitative and qualitative evidence requires greater clarity; and community perspectives are not well articulated. Addressing these issues (particularly refining scope, strengthening synthesis plans, and embedding cultural/community engagement) could substantially improve the rigor and utility of the review.

specific comments to consider:

Abstract: Clarify that this is a scoping review protocol, which maps interventions rather than formally evaluating effectiveness.

Introduction: Expand discussion of criminalization and legal barriers as critical contextual factors shaping intervention availability.

Methods: Justify exclusion of non-English studies and discuss potential language bias more explicitly.

Methods: Grey literature strategy is comprehensive; please clarify how program reports of variable quality will be assessed or contextualized.

Data Analysis: Provide more detail on how quantitative descriptive data and qualitative thematic analysis will be integrated into the final framework.

Limitations: Stronger emphasis on how structural stigma may limit published evidence, and how this will be interpreted in synthesis.

Dissemination: Consider including community dissemination via key population networks to validate findings and promote uptake.

Reviewers' comments:

Reviewer's Responses to Questions

**Comments to the Author**

1. Does the manuscript provide a valid rationale for the proposed study, with clearly identified and justified research questions?

Reviewer #1: Yes

2. Is the protocol technically sound and planned in a manner that will lead to a meaningful outcome and allow testing the stated hypotheses?

Reviewer #1: Yes

3. Is the methodology feasible and described in sufficient detail to allow the work to be replicable?

Reviewer #1: Yes

4. Have the authors described where all data underlying the findings will be made available when the study is complete?

Reviewer #1: Yes

5. Is the manuscript presented in an intelligible fashion and written in standard English?

Reviewer #1: Yes

You may also provide optional suggestions and comments to authors that they might find helpful in planning their study.

Reviewer #1: This is a clear, timely, and policy-relevant review protocol that addresses important gaps in HIV services for key populations. The rationale is strong, and using the JBI method adds rigor, while the inclusion of grey literature and plans to develop an integrated framework strengthen its relevance. However, some parts of the manuscript would benefit from clearer explanations, updating timelines and rephrasing.

**Do you want your identity to be public for this peer review?** For information about this choice, including consent withdrawal, please see our Privacy Policy

Reviewer #1: **Yes: ** Clara Ekpekose Oguji

---

## [Author Response · Author response to Decision Letter 1]

27 Sep 2025

Dear Editor

We would like to thank you and the reviewers for the thorough review of our manuscript and the insightful comments and suggestions offered for improving the overall quality of the work. We have revised the paper accordingly and believe the current version of the paper meets the journal’s publication requirements.

Please find below our responses to the editor and reviewer comments, detailing the changes we have made to the paper.

Editor Comments

General Comment: This protocol addresses a highly relevant global health issue by proposing a systematic scoping review of HIV prevention, treatment, and care interventions for key populations. The rationale is strong, the methodology is anchored in JBI and PRISMA-ScR guidelines, and the inclusion of both biomedical and regional databases enhances comprehensiveness. The strengths lie in its holistic framing and attention to structural barriers. However, there are notable limitations: the scope risks overreaching by conflating mapping with effectiveness assessment; restricting to English-only studies may introduce language bias also; the integration strategy for quantitative and qualitative evidence requires greater clarity; and community perspectives are not well articulated. Addressing these issues (particularly refining scope, strengthening synthesis plans, and embedding cultural/community engagement) could substantially improve the rigor and utility of the review.

Response: We sincerely thank you for the thoughtful and constructive feedback on our protocol. We have carefully considered all the concerns raised and revised the manuscript accordingly. The issues identified—clarifying the scope of the review as a mapping exercise rather than an effectiveness assessment, providing a stronger justification for the exclusion of non-English studies and explicitly acknowledging potential language bias, elaborating on the integration of quantitative and qualitative evidence, and embedding community dissemination through key population networks—have all been addressed.

Specific responses to each of these points are detailed below:

Comment 1: Abstract: Clarify that this is a scoping review protocol, which maps interventions rather than formally evaluating effectiveness.

Response: The abstract has been revised to explicitly state that this is a scoping review protocol, which aims to map the range of HIV interventions targeting key populations rather than formally evaluating their effectiveness.

Comment 2: Introduction: Expand discussion of criminalization and legal barriers as critical contextual factors shaping intervention availability.

Response: The introduction has been revised to expand the discussion of criminalization and legal barriers as critical contextual factors that shape the availability and accessibility of HIV interventions for key populations.

Comment 3: Methods: Justify exclusion of non-English studies and discuss potential language bias more explicitly.

Response: The methods section has been revised to provide justification for the exclusion of non-English studies, noting practical constraints such as limited translation resources and the need for consistency in data extraction. The revision also explicitly acknowledges the potential for language bias and states that this limitation will be considered when interpreting the findings.

Comment 4: Methods: Grey literature strategy is comprehensive; please clarify how program reports of variable quality will be assessed or contextualized.

Response: The section on evidence types has been revised to clarify how program reports and other grey literature of variable quality will be assessed and contextualized. Specifically, we now state that such documents will not be excluded a priori but will be carefully appraised and contextualized during analysis, with their contributions considered mainly in terms of providing practical insights and complementing peer-reviewed evidence. Limitations in methodological rigor or reporting quality will be explicitly acknowledged to ensure transparency in interpretation.

Comment 5: Data Analysis: Provide more detail on how quantitative descriptive data and qualitative thematic analysis will be integrated into the final framework.

Response: The section on data analysis and synthesis has been revised to provide more detail on how quantitative descriptive data and qualitative thematic analysis will be integrated. Specifically, we now explain that a convergent synthesis approach will be used to map quantitative patterns (e.g., coverage levels, outcome trends) alongside qualitative themes (e.g., barriers, facilitators, contextual factors), thereby linking statistical trends with explanatory insights. This integrated framework will more clearly demonstrate the combined impact of these findings on the success of HIV interventions for key populations.

Comment 6: Limitations: Stronger emphasis on how structural stigma may limit published evidence, and how this will be interpreted in synthesis.

Response: The limitations section has been revised to explicitly acknowledge how structural stigma and criminalization may constrain what is studied, published, or disseminated regarding key populations. We also clarify that this potential bias will be taken into account when interpreting the findings to ensure sensitivity to the ways stigma and structural barriers shape the evidence base.

Comment 7: Dissemination: Consider including community dissemination via key population networks to validate findings and promote uptake.

Response:

The dissemination plan has been revised to include engagement with key population networks to validate findings and promote uptake at the community level. This addition ensures that dissemination is not only academic and policy-oriented but also responsive to the needs and perspectives of affected communities

Reviewer 1

Comment 1

Title

The topic is relevant and important; however, it could benefit from revision:

1. Consider including a geographical focus; is this for the global community or Africa, or Europe?

2. Consider emphasizing the outcome of the review, “ A scoping review of interventions for key populations, rather than a Protocol.

3. This way we can clarify the “Why”.

Response

We thank you for these constructive suggestions. Based on the recommendation to clarify the geographical focus and emphasize the scope of the review, we have revised the title to read: “Global HIV Prevention, Treatment, and Care Interventions and Strategies for Key Populations: Protocol for a Scoping Review.” This revision highlights the global scope and specifies the focus on key populations. However, since the manuscript is a protocol for the actual scoping review, we have retained the term “Protocol” in the title to accurately reflect the nature of the work.

Comment 2

Abstract Introduction

Consider specifying who the key populations are early in the abstract/introduction for clarity. This will help readers immediately understand the target groups of interest.

Response

We appreciate this suggestion and have revised the Abstract Introduction to specify the key populations earlier for greater clarity.

Comment 3

Line 61, 62

This is good evidence. Can you consider the proportion of this 88 million that are key population? If available?

Response

Thank you for this insightful suggestion. Unfortunately, the available global data on the cumulative 88 million HIV infections does not provide a disaggregation that attributes a specific proportion of these cases to key populations. However, we do present evidence on the disproportionate burden of HIV among key populations using more recent and disaggregated prevalence and incidence data. For instance, the subsequent sections of the introduction highlight that prevalence among men who have sex with men, transgender people, people who inject drugs, and sex workers is several times higher than in the general population, and that more than half of all new infections in 2022 occurred among key populations and their partners. We believe this approach provides the clearest available evidence on the magnitude of HIV’s impact on key populations

Comment 4

Line 81- 84

I would suggest you move this up. I was waiting to see the definition of the Key population.

Response

We appreciate your suggestion to move the definition of key populations earlier in the introduction. However, we have deliberately structured the introduction to first provide a broader overview of the global HIV epidemic and the progress made in prevention and treatment, before narrowing the focus to the specific populations of interest. This progression is intended to set the stage for the reader by situating the discussion of key populations within the wider global context of the HIV response. We believe this structure enhances clarity and flow, guiding readers from the general epidemic landscape toward the rationale for focusing on key populations as the central theme of the review. For this reason, we respectfully maintain the current placement of the definition of key populations.

Comment 5

Line 123-124

Consider briefly justifying the choice of a scoping review over a systematic review. While the broad scope and numerous evidence sources align with this approach, you should explicitly state why a scoping review is more appropriate.

Response

We thank you for this valuable suggestion. The introduction has been revised to briefly justify why a scoping review is more appropriate than a systematic review, highlighting the broad scope and heterogeneity of the evidence base.

Comment 6

Line 132

Consider combining objectives 4 and 5 to avoid redundancy. Except you want the framework to be a standalone objective

Response

Thank you for the suggestion. Objectives 4 and 5 have been combined to avoid redundancy.

Comment 7

Line 161 -163

Are we considering any location for this? As mentioned in my previous comment.

Response

Thank you for this observation. The study considers the global community, and this has been specified in the revised title.

Comment 8

Line 167

Consider Rephrasing to “ We will search PubMed to identify…….

Response

Thank you for the suggestion. The phrasing has been corrected to “We will search PubMed to identify…”.

Comment 9

Line 272

Kindly review the timeline.

Response

We appreciate your observation. We commenced the literature search in June 2025, and considering the extensive nature of the search, we have revised the timeline accordingly: literature search beginning in June 2025, first draft expected by April 2026, and submission for publication anticipated in June 2026

Comment 10

Line 278 -279

In addition, it could limit the ability to make definitive practice recommendations.

Response

This has been added accordingly

Comment 11

Line 384

Double-checked for formatting, and kindly add the date assessed.

Response

Thank you for the observation. The reference has been checked for formatting and revised to include the date assessed

---

## [Decision Letter · Decision Letter 1]

4 Nov 2025

Global HIV Prevention, Treatment, and Care Interventions and Strategies for Key Populations: Protocol for a Scoping Review

PONE-D-25-06952R1

Dear Dr. Kumah,

We’re pleased to inform you that your manuscript has been judged scientifically suitable for publication and will be formally accepted for publication once it meets all outstanding technical requirements.

Kind regards,

Udoka Okpalauwaekwe, MD, MPH, PhD

Academic Editor

PLOS ONE

Additional Editor Comments (optional):

Thanks for addressing the comments for this piece. I look forward to reading your work in the future.

Reviewers' comments:

Reviewer's Responses to Questions

**Comments to the Author**

1. Does the manuscript provide a valid rationale for the proposed study, with clearly identified and justified research questions?

Reviewer #1: Yes

2. Is the protocol technically sound and planned in a manner that will lead to a meaningful outcome and allow testing the stated hypotheses?

Reviewer #1: Yes

3. Is the methodology feasible and described in sufficient detail to allow the work to be replicable?

Reviewer #1: Yes

4. Have the authors described where all data underlying the findings will be made available when the study is complete?

Reviewer #1: Yes

5. Is the manuscript presented in an intelligible fashion and written in standard English?

Reviewer #1: Yes

You may also provide optional suggestions and comments to authors that they might find helpful in planning their study.

Reviewer #1: Authors addressed all comments and initial concerns. Thank you for the opportunity to review this protocol. The topic is timely, relevant, and well aligned with current discussions in public health.

**Do you want your identity to be public for this peer review?** For information about this choice, including consent withdrawal, please see our Privacy Policy

Reviewer #1: **Yes: ** Clara Oguji

---

## [Editor Report · Acceptance letter]

PONE-D-25-06952R1

PLOS ONE

Dear Dr. Kumah,

I'm pleased to inform you that your manuscript has been deemed suitable for publication in PLOS ONE. Congratulations! Your manuscript is now being handed over to our production team.

Kind regards,

on behalf of

Dr. Udoka Okpalauwaekwe

Academic Editor

PLOS ONE